YOLOv8n-DDSW: an efficient fish target detection network for dense underwater scenes

Yi Jinwang
Han Wei 2222031246@stu.xmut.edu.cn
Lai Fangfei
School of Opto-Electronic and Communication Engineering, Xiamen University of Technology , Xiamen, Fujian , China
Somani Arun
Electronic publication date: 2025 Apr 7
Publication date: 2025
Volume: 11
Electronic Location ID: e2798
Received 2024 Nov 19; Accepted 2025 Mar 12
Copyright: © 2025 Yi et al.
Copyright year: 2025
Copyright holder: Yi et al.
License: This is an open access article distributed under the terms of the Creative Commons Attribution License, which permits unrestricted use, distribution, reproduction and adaptation in any medium and for any purpose provided that it is properly attributed. For attribution, the original author(s), title, publication source (PeerJ Computer Science) and either DOI or URL of the article must be cited.
License URL: https://creativecommons.org/licenses/by/4.0/

Keywords: Fish target detection, YOLOv8n, C2f-DCN, DPSE, Small detect

Funding: Natural Science Foundation of Xiamen, China 3502Z20227218 National Natural Science Foundation of China 61701422 This work was supported by the Natural Science Foundation of Xiamen, China, under Grant 3502Z20227218 and the National Natural Science Foundation of China under Grant 61701422. There was no additional external funding received for this study. The funders had no role in study design, data collection and analysis, decision to publish, or preparation of the manuscript.

==============================
Aquaculture is of great significance to economic development. It is assessed by manual periodic sampling traditionally, consumes workforce and material resources, and quickly leads to inadequate supervision, which results in substantial property losses. Fish target detection technology can effectively solve the issue of manual monitoring. However, a majority of current studies are based on ideal underwater environments and are inapplicable to complex underwater aquaculture scenarios. Therefore, the YOLOv8n-DDSW fish target detection algorithm was proposed in this article to resolve the detection difficulties resulting from fish occlusion, deformation and detail loss in complex intensive aquaculture scenarios. (1) The C2f-deformable convolutional network (DCN) module is proposed to take the place of the C2f module in the YOLOv8n backbone to raise the detection accuracy of irregular fish targets. (2) The dual-pooling squeeze-and-excitation (DPSE) attention mechanism is put forward and integrated into the YOLOv8n neck network to reinforce the features of the visible parts of the occluded fish target. (3) Small detection is introduced to make the network more capable of sensing small targets and improving recall. (4) Wise intersection over union (IOU) rather than the original loss function is used for improving the bounding box regression performance of the network. Training and testing are based on the publicly available Kaggle dataset. According to the experimental results, the mAP50, precision (P), recall (R) and mAP50-95 values of the improved algorithm are 3.9%, 3.7%, 6.1%, and 7.7% higher than those of the original YOLOv8n algorithm, respectively. Thus, the algorithm is effective in solving low detection accuracy in intensive aquaculture scenarios and theoretically supports the intelligent and modern development of fisheries.

Introduction

Statistics show that global aquaculture has exceeded 122 million tons in production, and its development is essential for promoting the world economy (FAO, 2024). Nevertheless, aquaculture requires close attention to various factors and regular assessments to reduce the risk of aquaculture losses. Traditional manual sampling is time-consuming, labor-intensive and prone to disturbing the ecological environment, which makes it challenging to meet the needs of modern aquaculture development (Arechavala-Lopez et al., 2022; Liu et al., 2021). Therefore, underwater fish target detection technology needs to be studied to promote aquaculture development in the direction of automation and intelligence.

Fewer related studies exist due to complex underwater environments, difficult detection and scarce fish dataset images. In the early days, most studies were conducted by manually extracting fish shape, color and texture, and other shallow features and then using classifiers for recognition and classification (Le & Xu, 2017; Ulutas & Ustubioglu, 2021; Badawi, 2022). From using a single shape feature for detection to combining multiple shallow features for detection and then improving classifiers, it can be seen that some progress has been made in traditional appearance-based feature detection methods (Strachan, Nesvadba & Allen, 1990; Larsen, Olafsdottir & Ersbøll, 2009; Tharwat et al., 2018). As they can only extract shallow information, problems such as single feature extraction and matching, poor adaptability, slow detection speed and low detection accuracy can be found. Consequently, these methods are ineffective in underwater applications and have more significant limitations (Jing & Lulu, 2019).

Today, deep learning technology is developing rapidly, breaks through the limitations of traditional object detection algorithms and becomes the mainstream algorithm of the present time (Amanullah et al., 2022; Zhu, 2022). With excellent feature extraction capabilities, convolutional neural networks (CNN) based on deep learning can be classified into two types. The first type is the two-stage object detection algorithm that is represented by region-based CNNs (RCNNs). It produces candidate regions first and then carries out classification and regression. The second type is the one-stage object detection algorithm that is represented by YOLO. It directly uses predefined candidate boxes to generate where the object is and which category the object belongs to Fiorin et al. (2011), Girshick et al. (2014), Redmon et al. (2016). In 2015, a fish recognition system based on Fast-RCNN was proposed (Girshick, 2015). Subsequently, the Faster R-CNN algorithm was proposed, which improved detection speed by integrating the critical steps of two-stage object detection (Ren et al., 2017). In 2021, the Faster R-CNN algorithm was improved and applied to detect dense marine life (Liu & Wang, 2022). Two-stage object detection algorithms possess high detection accuracy, but their network structures are complex, which requires high hardware costs. The detection speed fails to satisfy real-time requirements. As a single-stage object detection algorithm, the YOLO model features a lighter network structure, higher detection speed and high detection accuracy. In 2022, YOLO-Fish-1, a fish detection model based on YOLOv3, was proposed, which improved detection accuracy by modifying the upsampling step. To enhance the detection ability of dynamic fish bodies, YOLO-Fish-2 added a spatial pyramid structure, which further improved detection accuracy (Muksit et al., 2022). With the development of the YOLO series of algorithms, its application in the detection of underwater fish targets has also become increasingly in-depth. In 2022, the detection accuracy of deformed fish targets was increased by introducing deformable convolutions to the YOLOv4 algorithm model (Zhao et al., 2022). In 2024, the YOLO-Waternet object detection model was proposed, which solved the problem of target occlusion but gave no consideration to the deformation of the target (Liu, Qian & Wang, 2024). In 2024, the YOLOv7 algorithm model was introduced with the normalization-based attention module (NAM) attention mechanism to suppress unimportant features in the image, which achieved an accuracy rate of 95% (Cai et al., 2024). With the popularity of the YOLOv8 network model, it has gradually been applied to underwater target detection scenarios. However, the model has certain limitations. When it is directly applied to dense underwater detection scenarios, detection accuracy is low. Therefore, it is necessary to conduct more research to apply the YOLOv8 network model to underwater detection (Zhou et al., 2024). The YOLOv8-LA model was proposed for the detection of underwater small targets, which enhanced the ability to capture small targets but failed to consider the deformation of targets (Qu et al., 2024). Subsequently, the BSSFISH-YOLOv8 model was proposed, which improved the detection accuracy of underwater targets by introducing an attention mechanism and a small target detection layer. However, it introduced some parameters for the attention mechanism, which resulted in poor real-time performance (Zhang et al., 2024). Thus, further improvement is needed to raise the detection accuracy of underwater fish targets while ensuring real-time performance.

In summary, current research is mostly based on conventional underwater scenes or static fish target detection and is inapplicable to complex underwater intensive aquaculture scenes. Detection is difficult owing to the complex underwater environment, easily missing small targets, the mutual blockade of fish bodies and the deformation of fish targets. To address these problems, the YOLOv8n-DDSW object detection model is proposed: (1) The C2f-deformable convolutional network (DCN) module is used in the YOLOv8n backbone to enhance the ability of the model to recognize deformed fish bodies. (2) Feature extraction capabilities are improved by introducing the dual-pooling squeeze-and-excitation (DPSE) attention mechanism into the neck of YOLOv8n. (3) The detection ability of small objects is enhanced via the introduction of a small object detection head. (4) Wise-intersection over union (IOU) improves the bounding box regression performance of the network. Experiments show that this algorithm effectively solves high missed detection rates and low detection accuracy in intensive farming scenarios.

Efficient fish target detection network design

YOLOv8 object detection model

The YOLOv8 detection model boasts high efficiency, fast detection speed and flexible target recognition. It can effectively detect a variety of targets and support multi-class classification tasks. The model is mainly composed of four modules, namely input, backbone, neck and head. The input end inputs features of different scales; the backbone end extracts input features; the neck fuses the extracted features and passes them to the detection head for target detection. Figure 1 displays the architecture of the YOLOv8 network.

Figure 1 YOLOv8n network structure.

Improved YOLOv8n-DDSW recognition method

Despite owning many advantages, the YOLOv8 model still has problems like missed detections and low detection accuracy when directly applied to complex underwater target detection scenarios. Hence, the YOLOv8n-DDSW model suitable for detecting fish targets in intensive aquaculture underwater is proposed. To better detect irregular fish targets, C2f-DCN is proposed and replaces C2f in the backbone. To enhance the features of occluded fish bodies, the DPSE attention mechanism is proposed and integrated into the neck network. Small detection is added to strengthen the perception of small targets by the network. Wise-IOU instead of the original loss function is utilized for improving the bounding box regression performance of the network. The model is shown in Fig. 2.

Figure 2 YOLOv8n-DDSW network structure.

C2f-DCN module

In the backbone of YOLOv8, feature extraction is primarily performed by the C2f module. However, underwater fish targets have substantial flexibility and morphological variability. Standard convolutions are only suitable for the detection of some static targets with regular shapes and low complexity. As a result, it is proposed that two DCN layers and one Deformable region of interest (ROI) pooling layer be introduced into the bottleneck structure of C2f. DCN can help the model better learn information such as the scale, background and deformation of a target. Deformable ROI pooling can learn offsets, expand the perception field of the model, and extract more complex features. The C2f-DCN module proposed in this article can adaptively learn targets of different sizes, better capture the features of the target, and show great robustness. The C2f-DCN structure is presented in Fig. 3.

Figure 3 C2f-DCN structure.

DCN module

Conventional standard convolutions are arranged in a regular grid and sampled at regular grid positions in a feature map. In the DCN, however, the convolution kernel is not a fixed N × N grid (Dai et al., 2017). Sampling positions are no longer limited to regular grid positions. Instead, free sampling is performed by introducing offsets, which thereby expands the perception field, as shown in Fig. 4.

Figure 4 Sampling process.

(A) Standard convolutional sampling points. (B) Deformable convolutional sampling points.

The standard convolution operation is generally divided into two steps: sampling the input feature map with a fixed grid and performing a weighting operation on the values of each sampling point. Its feature matrix formula is demonstrated in Eq. (1).

(1) y(p0)=∑k=1Kwk⋅x(p0+pk).

DCN adds an offset to the standard convolution, as shown in Eq. (2).

(2) y(p0)=∑k=1Kwk⋅x(p0+pk+Δpk)

where x and y represent input and output feature maps, respectively; K and k stand for the total number of sampling points and enumerated sampling points, respectively; p0 refers to the current pixel; wk denotes the projection weight of the kth sampling point; pk is the kth position sampled by the predefined convolution grid; and Δpk means the offset that corresponds to the kth sampling position.

In Eq. (2), the pixel x(p0+pk+Δpk) at the sampled position after the offset is usually implemented using a bilinear interpolation method.

(3) x(p)=∑qG(q,p)⋅x(q)

(4) G(q,p)=g(qx,px)g(qy,py)

(5) g(a,b)=max{0,1−|a−b|}

where p is an arbitrary position in the region, p=p0+pk+Δpk; q represents all the spatial positions in input feature mapping; x(q) stands for the value of all the integer points on the feature map; G denotes a bilinear interpolation kernel function, which is a two-dimensional kernel that can be decomposed into two one-dimensional kernels.

The specific implementation process of deformable convolution is illustrated in Fig. 5. The left and right sides of the picture represent input and output features, respectively. First, the convolution preprocesses input features to generate offsets and modulations. After that, these offsets transform regularly distributed 3 × 3-pixel points into irregular, arbitrarily distributed 9-pixel ones. Next, the input image is sampled using the generated new pixel points to get the sampled feature map. Finally, the sampled feature map is multiplied with the convolution kernel element by element and summed to obtain the final convolution result.

Figure 5 Deformable convolution process.

Deformable ROI pooling module

The shape and size of the deformable ROI pooling window are dynamically adjusted according to the target shape and size for better adaptation. Its process is displayed in Fig. 6. First, the ROI is automatically generated and divided into t × t subregions. Subsequently, the offset of each subregion is calculated, followed by the adjustment of the sampling position according to the offset to obtain the feature value of the new subregion. Next, adaptive maximum pooling is performed on the sampled feature values of each new subregion to obtain a fixed-size output. At last, the pooling results of all new subregions are spliced in a particular order to obtain the final pooling result.

Figure 6 Deformable ROI pooling process.

DPSE attention mechanism module

With the help of attention mechanisms, models can focus on targets and ignore background information, which improves their detection performance. Among attention mechanisms, the squeeze-and-excitation (SE) attention mechanism can adaptively adjust feature weights at the channel level to establish the interrelationship between channels and enhance critical features (Hu, Shen & Sun, 2018). Its structure is displayed in Fig. 7, where X and X′ represent input and output feature maps, respectively; C stands for the number of channels of the input feature map; H and W denote the height and width of the map, respectively. It mainly comprises two operations: squeeze and excitation. First, the squeeze operation employs global average pooling for converting the H × W feature layer into a 1 × 1 feature layer to obtain channel-level global features and the weights of each channel. Then, the excitation operation passes the 1 × 1 feature layer through two fully connected layers and normalizes it by use of sigmoid contrast weights to obtain channel weights. In the end, the feature map is used to multiply the output weight vector to get a feature map with weight information.

Figure 7 SE attention mechanism module.

The squeeze operation takes advantage of global average pooling, which is formalized, as shown in Eq. (6):

(6) zc=Fsq(uc)=1H×W∑i=1H∑j=1Wuc(i,j)

where zc represents the feature vector after the cth channel compression operation, Fsq stands for compression operation, and uc denotes the channel of the input feature map. uc(i,j) refers to the feature value of the ith row and the jth column of the cth channel of the input feature map; H and W indicate the height and width of the feature map, respectively.

The excitation operation is shown in Eq. (7):

(7) SC=Fex(zc,W)=σ(W2δ(W1zc))

where, sc represents the feature weight of the cth channel after the excitation operation; Fex stands for excitation operation; W1, W2 indicate the weight parameters of the two fully-connected layers; σ denotes the Sigmoid activation function; and δ refers to the ReLU activation function.

The channel recalibration operation is shown in Eq. (8):

(8) x~c=Fscale(uc,sc)=scuc

where, X~=[x~1,x~2,…,x~C],uc∈RH×W.Fscale(uc,sc) represents channel-wise multiplication between sc and uc. sc and uc denote the scalar and feature map, respectively.

The SE attention mechanism applies global average pooling, which averages the feature values of all channels and can retain more channel information. Nonetheless, average pooling causes the model to lose some essential local feature information, while max pooling can highlight critical local features. For this reason, the DPSE attention mechanism, which makes a combination of average and max pooling, is proposed. Average pooling highlights background information and outputs the average value of activations in the image, whereas maximum pooling highlights texture information and outputs the maximum value of activations. This model can pay attention to global and local information simultaneously, which thereby improves its performance and robustness. Figure 8 presents the structure of the DPSE attention mechanism.

Figure 8 DPSE attention mechanism module.

Equations (9) and (10) show the forms of average pooling and maximum pooling, respectively.

(9) X¯C=1H×W∑i=1H∑j=1Wuc(i,j)

(10) Maxc=Maxuc(i,j),(1≤i≤H,1≤j≤W).

The squeeze operation is formalized, as shown in Eq. (11):

(11) zc=Fsq(uc)=X¯C+Maxc.

The excitation operation is shown in Eq. (12). A Sigmoid activation function is added after the double pooling operation to capture the result of double pooling.

(12) sc=Fex(zc,W)=σ(W2δ(W1(σzc))).

The channel recalibration operation is formalized, as shown in Eq. (13).

(13) x~c=Fscale(uc,sc)=scuc.

The DPSE attention mechanism proposed in this article combines the merits of average and maximum pooling. This model retains more channel-level information while increasing attention to essential features, which leads to a better understanding of the correlation and importance between different regions.

Small detection module

Small object detection has always been a challenge because losing the characteristic information of small objects is easy. The original YOLOv8 algorithm has three detection heads detecting small, medium and large targets. The input size is a 640 × 640 feature map. After several downsampling operations, the feature map at the detection layers has a size of 20 × 20, 40 × 40 and 80 × 80, respectively. These sizes are utilized for detecting large targets above 32 × 32, medium targets above 16 × 16 and small targets above 8 × 8, respectively. However, the detection head of YOLOv8 still has limited detection capabilities for small targets. It fails to make full use of shallow feature maps with richer information about small targets. In some cases, small targets are not detected in practical underwater scene applications.

Therefore, a small target detection layer with a 160 × 160 feature map is added to detect targets above 4 × 4. Upsampling and feature stitching are performed again based on the original algorithm model to obtain more detailed feature information on small targets. Subsequently, this feature information keeps being passed to the other three scale feature layers along the path of downsampling. The small detection head not only enhances the ability of the network to fuse features but also provides more shallow feature and positioning information. The structure is illustrated in Fig. 9.

Figure 9 Small detect structure.

Wise-IOU

IOU-based loss functions are widely used in object detection tasks. YOLOv8 uses complete IOU (CIOU) for calculating the regression loss of the bounding box. This loss function evaluates the resemblance between two bounding boxes based on the position, size and angle differences between them and calculates positioning loss, as shown in Eq. (14).

(14) LClOU=1−IOU+ρ2(bA,bB)c2+αν

where bA and bB represent the centroids of prediction and real frames, respectively, ρ stands for the Euclidean distance between both points, and c denotes the diagonal length of the smallest outer rectangle of prediction and real frames; α refers to a balancing parameter. In addition, v is used for computing the consistency of the aspect ratios of prediction and target frames.

However, CIOU uses a monotonic focusing mechanism that rules out the balance between simple and complex samples. When training samples contain low-quality targets, the detection performance of the model decreases. Therefore, this article introduces the Wise-IOU loss function with a dynamic non-monotonic focusing mechanism for balancing the samples, as shown in Eqs. (15) to (17).

(15) LWIOU=rRWIOULIOU,r=βδαβ−α

(16) β=LIOU∗LIOU¯∈[0,+∞)

(17) RWIOU=exp⁡((x−xgt)2+(y−ygt)2(cw2+ch2)∗)

where LIoU∈[0,1] represents the IOU loss attenuating the penalty term for high-quality anchor frames; RWIOU∈[0,exp] stands for the Wise-IOU penalty term strengthening the loss of normal-quality anchor frames; the superscript * represents not participating in backpropagation; LIOU¯ is a normalization factor standing for the sliding average of increments; β refers to the degree of outliers, where smaller values imply higher-quality anchor frames.

Dataset

The dataset used in this experiment consists of 2,450 images, which include five types of underwater scenes, namely the mutual occlusion of fish bodies, the deformation of fish bodies, occlusion by aquatic plants, blurred light and the loss of details in numerous small targets. Sourced from the Kaggle platform, one underwater scene contains 629 underwater fish images and is available for research under the Kaggle Open Data License. The other four underwater scenes are from the DeepFish public dataset and comprise 407, 450, 502 and 462 images, respectively. The dataset is divided into training, validation and test sets, with proportions of 70%, 15% and 15%, respectively. A sample of the dataset is presented in Fig. 10.

Figure 10 (A–E) Sample plot of the dataset.

Experimental protocols and evaluation measures

Experimental platform and protocols

The platform for this experiment is the Windows 11 operating system. The hardware device has a memory of 16.0 GB. The GPU is NVIDIA RTX4060. The software environment is Python 3.7 based on the PyTorch2.0.0 framework and configured with CUDA 12.0 for computational acceleration. YOLOv8n was used as the baseline network model with an initial learning rate of 0.01, a final learning rate of 0.01 and an SGD optimizer. The number of training rounds is set to 130. The training batch size is 8, and the input image size is 640 * 640.

Model evaluation measures

In this article, precision (P), recall (R), mean average precision 50 (mAP50) and mAP50-95 were taken as assessment metrics. The following parameters were used in the formulas of the above assessment indexes: true positive (TP) represents the case in which the actual category is positive and also predicted to be positive; false negative (FN) indicates the case in which the actual category is positive but predicted to be negative; false positive (FP) refers to the case in which the actual category is harmful but predicted to be positive; true negative (TN) denotes the case in which the actual category is harmful and predicted to be negative, as shown in Table 1.

Table 1 Division of positive and negative samples.

Real value	Predicted value	
Positive	Negative	
Positive	TP	FN	
Negative	FP	TN	

Precision represents the ratio of the number of positive samples predicted by the model to that of all the detected samples, as illustrated in Eq. (18):

(18) Precision=TPTP+FP

Recall denotes the ratio of the number of positive samples predicted by the model correctly to that of actually occurring positive samples, as illustrated in Eq. (19):

(19) Recall=TPTP+FN

AP is equal to the area under the precision-recall (PR) curve, as illustrated in Eq. (20):

(20) AP=∫01Precision(Recall)d(Recall)

Obtained by a weighted average of AP values for all sample categories, mAP is used for measuring the detection performance of the model across all categories, as illustrated in Eq. (21):

(21) mAP=1N∑i=1NAPi

where N represents the number of classes of dataset samples; APi stands for the AP value of class i; mAP50 and mAP50-95 denote average accuracy when the IOU is set as 0.5 and 0.5–0.95, respectively.

Results and discussion

Results analysis

After 130 rounds of training, the model converged and achieved good results on both training and validation sets. Additionally, box_loss, cls_loss and dfl_loss denote anchor box, classification and assigned focus losses, respectively, and smaller values of these losses indicate more accurate classification. As the results, loss values decrease significantly with the increasing number of training batches, and the values of accuracy, recall, mAP50 and mAP50-95 experience a significant increase and level off towards the end of training. Figure 11 displays the training results of the YOLOv8n-DDSW model.

Figure 11 Training results of YOLOv8n-DDSW model.

The PR curve is a standard metric for evaluating model performance. It takes recall and precision as horizontal and vertical axes, respectively and reflects the change in precision under different recall rates. Model performance under different recall rates is observable intuitively. The model has higher accuracy and recall when the curve is closer to the upper right corner, which indicates that the model has good performance. Figure 12 demonstrates the PR curves of YOLOv8n and YOLOv8n-DDSW models. The PR curve of the YOLOv8n-DDSW model developed is closer to the upper right corner and has better detection performance.

Figure 12 (A, B) PR curve results.

To demonstrate the detection performance of the model in this article, representative data were selected for visualization experiments. The experimental results are illustrated in Fig. 13. In Scene 1, the fish targets are densely packed, which result in significant occlusion and deformation. In Scene 2, the fish bodies appear blurred and deformed, with incomplete features. In Scene 3, the fish bodies are obscured by aquatic plants. This causes the loss of small target details. The yellow wireframe area in the figure indicates instances of misdetection and omission by the YOLOv8n algorithm. In contrast, these targets are successfully detected by the YOLOv8n-DDSW algorithm. These results clearly show that the proposed algorithm greatly improves the ability of the model to detect fish targets in various underwater scenarios.

Figure 13 Comparison of model detection effects.

(A–C) YOLOv8n model detection results. (D–F) YOLOv8n-DDSW model detection results.

Ablation experiment

To evaluate the effectiveness of the proposed algorithm, six ablation experiments were performed using the same experimental data set under the same experimental conditions. The performance impact of DCN, DPSE, small detection and Wise-IOU on the YOLOv8n model was explored separately. Table 2 presents the results of objective evaluation metrics.

Table 2 Impact of each improvement module on detection performance.

Number	Model	mAP50 (%)	P (%)	R (%)	mAP50-95 (%)	
(1)	YOLOv8n	93.4	91.9	86.3	69.3	
(2)	YOLOv8n-DCN-DPSE	96.4	94.6	89.5	73.8	
(3)	YOLOv8n-DPSE-Small detect	96.3	94.1	90.2	73.9	
(4)	YOLOv8n-DCN-Small detect	96.4	93.8	91.3	74.2	
(5)	YOLOv8n-DCN-DPSE-Small detect	97.0	95.1	91.8	75.6	
(6)	YOLOv8n-DDSW	97.3	95.6	92.4	77.0	

The experimental results in the table above indicate that mAP50, P, R and mAP50-95 reach 97.0%, 95.1%, 91.8%, and 75.6%, respectively when the DCN, DPSE and small-detection modules are used together. When one of these three modules is removed, the model experiences a decrease in detection performance: (1) After the removal of the DCN module, the P and R of the model decrease by 1.0% and 1.6%, respectively, which suggests that the DCN module is capable of improving the low detection accuracy and missed detection caused by fish deformation. (2) After the removal of the DPSE module, P decreases by 1.3%, which indicates that the DPSE module can improve the low detection accuracy resulting from the mutual occlusion of fish bodies. (3) When small detections are removed, the R-value decreases by 2.3%, which indicates that this module can solve the problem of missed detection of small objects and raise the recall rate of the model. When the four proposed modules are used together, the detection effect is optimal, with mAP50, P, R and mAP50-95 values reaching 97.3%, 95.6%, 92.4% and 77.0%, respectively, which suggests that the loss function improved optimizes the model better. Figure 14 compares the mAP50 and R values in the six ablation experiments.

Figure 14 (A, B) Comparison of the ablation experiment results.

Comparative experiment

Comparative experiment of attentional mechanisms

Different attention mechanisms were embedded into the network model for comparison, to verify the effectiveness of the DPSE attention mechanism. The results are shown in Table 3. The first group is the control group that used the original YOLOv8n algorithm. From the second to the sixth group, squeeze and excitation (SE), efficient channel attention (ECA), convolutional block attention module (CBAM), Biformer and DPSE mechanisms were added to YOLOv8n, respectively. The results show that the mAP50 values of SE and DPSE mechanisms increased by 1.1% and 1.9%, respectively, without increasing parameters. The ECA mechanism increased parameters slightly and mAP50 by 1.4%. The CBAM mechanism significantly increased parameters, but only improved mAP50 by 0.4%. The Biformer mechanism led to a substantial increase in parameters and a decrease in accuracy. It can be seen from these results that the DPSE attention mechanism proposed in this article has the best performance.

Table 3 Impact of different attention mechanisms on the detection performance of models.

Group	Model	mAP50 (%)	P (%)	R (%)	Parameters/106	
(1)	YOLOv8n	93.4	91.9	86.3	3.27	
(2)	YOLOv8n-SE	94.5	92.7	88.7	3.27	
(3)	YOLOv8n-ECA	94.8	93.0	89.1	3.28	
(4)	YOLOv8n-CBAM	93.8	92.1	86.5	3.36	
(5)	YOLOv8n-Biformer	91.7	90.3	85.8	4.18	
(6)	YOLOV8n-DPSE	95.3	93.6	90.7	3.27	

Comparison experiment of replacing different number of DCN

To verify the effect of DCNs on network performance, how replacing different numbers of DCNs in the bottleneck structure of C2f influences performance was investigated. The bottleneck structure includes two convolutional layers. Five groups were designed for this experiment. Group 1 serves as the control group and is composed of the original YOLOv8n network model. Groups 2 and 3 are based on the YOLOv8n model and take the place of the 1- and 2-layer convolution in the bottleneck, respectively. Group 4 is based on the model proposed in this article and replaces the 1-layer convolution. Group 5 is the model proposed in this article and replaces the 2-layer convolution. The experimental results are presented in Table 4.

Table 4 Comparison of replacing different numbers of DCNs.

Group	mAP50 (%)	P (%)	R (%)	GFLOPs	Parameters/106	
(1)	93.4	91.9	86.3	8.1	3.27	
(2)	93.8	92.4	87.2	7.9	3.18	
(3)	94.3	93.6	88.5	7.5	3.12	
(4)	96.6	94.8	91.7	8.0	3.21	
(5)	97.3	95.6	92.4	7.7	3.14	

As demonstrated in Table 4, both parameters and GFLOPs decrease with the increasing number of replaced DCNs. Compared to the model with only one layer of convolutional substitution, the model proposed in this article improves mAP50, precision (P) and recall (R) values by 0.7%, 0.8% and 0.7%, respectively. Consequently, this article replaces all two layers of standard convolution at the bottleneck with DCNs.

Comparative experiment of loss function

To verify whether the loss function improved is effective, comparative experiments were performed based on the improved model YOLOv8n-DCN-DPSE-Small detection proposed. The experimental results are shown in Eqs. (5) and (6) in Table 2. The regression loss in the process of training is presented in Fig. 15. It can be found that the loss function optimized converges faster, and the predicted value gets closer to the actual one.

Figure 15 Loss curve graph before and after improvement.

Comparative experiment of different detection algorithms

The results in Table 5 indicate that the two-stage target detection algorithm Faster-RCNN has significantly larger parameters and GFLOPs but lower detection accuracy than single-stage algorithms. The YOLOv8n-DDSW algorithm proposed in this article outperforms single-stage algorithms such as selective synaptic dampening (SSD), YOLOv5s, YOLOv7 and YOLOv7-tiny in terms of detection accuracy and the calculated number of parameters. Compared to the original YOLOv8n algorithm, the YOLOv8n-DDSW algorithm improves mAP50, precision (P) and recall (R) values by 3.9%, 3.7% and 6.1%, respectively, while slightly reducing the number of parameters and computations. Comprehensive analysis shows that the algorithm presented in this article surpasses current mainstream target detection algorithms and possesses significant advantages in detecting underwater fish targets.

Table 5 Comparison of detection performance among different models.

Group	Model	mAP50 (%)	P (%)	R (%)	GFLOPs	Parameters/106	
(1)	Faster R-CNN	89.6	87.3	84.5	206.7	41.21	
(2)	SSD	87.1	84.9	82.7	61.3	24.28	
(3)	YOLOv5s	93.1	91.8	88.4	15.8	7.14	
(4)	YOLOv7	92.7	91.8	87.5	105.3	37.21	
(5)	YOLOv7-tiny	93.2	91.9	88.7	13.2	6.28	
(6)	YOLOv8n	93.4	91.9	86.3	8.1	3.27	
(7)	YOLOv8n-DDSW	97.3	95.6	92.4	7.7	3.14	

Reasoning experiments

To verify the generalization of the proposed model, inference experiments were performed using another publicly available underwater fish dataset (Ditria et al., 2021). This dataset features a complex environment with plenty of small targets and occlusions from aquatic plants, which presents a challenge for inference experiments. The visualization results are shown in Fig. 16. The yellow wireframe part in the figure means that the YOLOv8n algorithm has missed detection and misdetection, which however can be detected by the YOLOv8n-DDSW algorithm in this article.

Figure 16 (A–D) Visualization results of inference experiments.

The experimental results demonstrate that the proposed YOLOv8n-DDSW algorithm improves mAP50, P and R values by 3.0%, 3.3%, and 5.5%, respectively on this dataset compared with the original YOLOv8n model. This suggests that the proposed algorithm is more robust. The results are shown in Table 6.

Table 6 Comparison of detection performance on public datasets.

Model	mAP50 (%)	P (%)	R (%)	
Faster R-CNN	88.9	85.1	80.2	
SSD	88.3	82.9	79.7	
YOLOv7-tiny	92.8	89.7	86.5	
YOLOv8n	94.7	92.4	87.3	
YOLOv8n-DDSW	97.7	95.7	92.8	

Conclusions

Some challenges still exist when the YOLOv8n model is directly applied to detection in underwater intensive aquaculture scenarios. Therefore, the YOLOv8n-DDSW fish object detection algorithm was proposed in this article to cope with these challenges. Firstly, the C2f-DCN module is proposed and applied to the backbone network to enhance the detection capabilities of fish body targets of different sizes and shapes. Secondly, the DPSE attention mechanism is proposed for enhancing the features covering the visible parts of fish bodies. Then, the detection ability of small target fish is enhanced by adding a small target detection head. Finally, the Wise-IOU rather than CIOU loss function is adopted to improve the bounding box regression performance of the network. Experiments show that the proposed algorithm model has higher detection accuracy than some current mainstream target detection models and can effectively solve low detection accuracy and high missed detection rates in underwater intensive aquaculture scenes. Despite being able to theoretically support the development of fisheries, the model proposed in this article is not allowed to learn increasingly complex underwater scenes due to the current scarcity of underwater fish datasets. Because the acquisition of underwater images requires high-tech equipment and shows susceptibility to a variety of underwater disturbances, most underwater target datasets are collected in more ideal environments. Therefore, future studies will collect more relevant underwater fish datasets, explore more realistic and complex underwater scenes, solve other technical problems facing the fishing industry, and continuously improve the generalization performance of the model.

Additional Information and Declarations

Competing Interests

The authors declare that they have no competing interests.

Author Contributions

Jinwang Yi conceived and designed the experiments, performed the experiments, analyzed the data, performed the computation work, prepared figures and/or tables, authored or reviewed drafts of the article, and approved the final draft.

Wei Han conceived and designed the experiments, performed the experiments, analyzed the data, performed the computation work, prepared figures and/or tables, authored or reviewed drafts of the article, and approved the final draft.

Fangfei Lai analyzed the data, prepared figures and/or tables, and approved the final draft.

Data Availability

The following information was supplied regarding data availability:

The original dataset is available at Zenodo: Yi, J., Han, W., & Lai, F. (2025). YOLOv8n-DDSW dataset [Data set]. Zenodo. https://doi.org/10.5281/zenodo.14903554.

Additional information is available at Zenodo: Yi, J., Han, W., & Lai, F. (2025). YOLOv8n-DDSW: an efficient fish target detection network for dense underwater scenes (V1.0.0). Zenodo. https://doi.org/10.5281/zenodo.14903287.

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
