# Peer review of "YOLOv8n-DDSW: an efficient fish target detection network for dense underwater scenes"

_PeerJ Computer Science, doi:10.7717/peerj-cs.2798_

## Round 0.1 · original submission · Major Revisions

Several reviewers have now commented. Overall, you should be sure to address the comments of Reviewers 3 and 4 in detail

Reviewer 1 ·

Basic reporting

Manuscript is well written using scientific English language. Objectives and methodology are clearly defined. Background and literature survey is adequate. References used are from relevant indexed journals and conferences. Script is supported by neatly drawn figures and result tables.

Experimental design

It indicates original research.
(1) The C2f module in the YOLOv8n backbone is replaced by C2f-deformable convolutional network (DCN) module to raise the detection accuracy of irregular fish targets.
(2) The Dual-pooling squeeze-and-excitation (DPSE) attention mechanism is put forward and integrated into the YOLOv8n neck network to reinforce the features of the visible parts of the occluded fish target.
(3) Small detection is introduced to make the network more capable of sensing small targets and improving recall.
(4) To improve the bounding box regression performance Wise intersection over union (IOU) rather than the original loss function is used.

Experimentation is done by individual addition of attention mechanisms, loss function and different detection algorithms.

Validity of the findings

One major contribution by authors is YOLOv8n-DDSW target detection algorithm to resolve the detection difficulties resulting from fish occlusion, deformation and detail loss in complex intensive aquaculture scenarios.
All underlying data have been provided; they are robust, statistically sound, & controlled.
Results, Discussions and conclusions are well stated.

Additional comments

Overall it’s a good work and contributes to the field of automation in underwater object detection.

Reviewer 2 ·

Basic reporting

Well written and easy to follow generally.

Literature review/introduction is too short, need to be enhanced with more references on YOLOv8 modification and applications analyses.

Experimental design

Suitable experiments were conducted.

Validity of the findings

It would be more interesting to see application in another similar datasets in fisheries for results robustness evaluation of the proposed algorithm.

Additional comments

There should be additional metrics to see the real-time performance of the model such as flops/parameter count and fps assessment. this can be added in Table 4 or new table.

Reviewer 3 ·

Basic reporting

The manuscript proposes YOLOv8n-DDSW, an improved version of the YOLOv8 network tailored for fish target detection in dense underwater environments. The authors introduce several modifications, including deformable convolutions, a dual-pooling squeeze-and-excitation (DPSE) attention mechanism, a small target detection head, and a new Wise-IOU loss function. The work addresses an practical issue in aquaculture monitoring, but several aspects of the study need significant improvement.

Experimental design

1. Dataset Scale and Selection. The dataset used in this study contains only 629 images, which is insufficient to assess the model's robustness and generalizability. Additionally, the dataset focuses on a single species in a single environment. Consider incorporating additional public datasets related to fish detection, such as DeepFish, or expanding the dataset to include multiple species and environments to make the study more comprehensive.

2. Dataset Split Issues. The current dataset is divided into training, validation, and test sets in a 7:2:1 ratio, leaving only ~60 images for testing. This is insufficient to evaluate the model’s generalization in real-world scenarios. I suggest adopting a spatially stratified sampling strategy or ensuring a larger test set for meaningful evaluation.

3. Limited Baseline Comparisons The comparison is limited to YOLOv8 and its variants. The manuscript would be significantly strengthened by including two-stage object detection methods like Faster R-CNN and SSD, as well as other state-of-the-art one-stage detectors.

4. Resource Efficiency Evaluation. While the authors highlight improvements in accuracy, there is no discussion of computational efficiency. Considering the additional modules introduced (e.g., deformable convolutions and DPSE), the paper should report the parameter count, inference time, training time, and GPU memory usage for all compared methods.

Validity of the findings

no comment

Additional comments

1. Motivation for Choosing YOLO-Based Models. The authors justify the choice of YOLO by emphasizing speed. However, they do not explicitly state whether real-time performance is a critical requirement for the task. If not, explain why YOLO was preferred over two-stage methods, which generally offer better accuracy.

2. Justification for Additional Modules. The manuscript introduces several modules, such as deformable convolutions and DPSE, without providing sufficient motivation. Why were these specific modules selected? I recommend visualizing feature maps before and after applying these modules to show their contribution to improved detection performance.

3. Literature Review. The review of related work is incomplete. Recent advances in underwater detection and remote sensing, such as those using Mamba architectures or specific methods for underwater imagery enhancement, are not discussed, including [0.1109/TGRS.2025.3526927], [10.1109/TGRS.2024.3417253], and [10.1109/DTPI61353.2024.10778881].

Since underwater imagery can also be considered remote sensing, it would be beneficial to include insights from general remote sensing literature.

Reviewer 4 ·

Basic reporting

This paper introduces a efficient fish target detection network for dense underwater scenes. Many experiments are used to verify the effectiveness of the proposed method. The following points could be considered.

The motivation and novelty in the abstract and Introduction should be improved.

In section 2, the C2f-DCN module needs more detailed theoretical analysis and mathematical derivation. More ablation studies are needed to justify the specific design choices of the DPSE attention mechanism.

Experimental design

The dataset size (629 images) is insufficient for deep learning model training. More details about data augmentation techniques are needed.

The comparison with state-of-the-art methods is limited. More recent fish detection algorithms in comparisons are needed. Comparison of computational costs and inference times are suggested to be included.

Analysis of model performance under different underwater conditions is missing. Performance on varying fish sizes and orientations should be analyzed. Impact of water quality on detection performance is not analyzed.

More qualitative results showing detection performance are needed.

Validity of the findings

Current limitations and real-world application constraints are not adequately discussed.

Training protocol needs more detailed description. Inference optimization strategies should be discussed. More visual results are needed.

---

## Round 0.2 · accepted · Accept

Thanks for your interest in the journal.

Reviewer 3 ·

Basic reporting

all of my concerns have been addressed.

Experimental design

no comment

Validity of the findings

no comment

Reviewer 4 ·

Basic reporting

All my concerns have been addressed. I recommend the paper for publication.

Experimental design

All my concerns have been addressed. I recommend the paper for publication.

Validity of the findings

All my concerns have been addressed. I recommend the paper for publication.

Additional comments

All my concerns have been addressed. I recommend the paper for publication.